# Extraction, Isolation and Characterization of Bioactive Compounds from *Artemisia* and Their Biological Significance: A Review

**DOI:** 10.3390/molecules26226995

**Published:** 2021-11-19

**Authors:** Rosemary Anibogwu, Karl De Jesus, Samjhana Pradhan, Srinath Pashikanti, Sameena Mateen, Kavita Sharma

**Affiliations:** 1Department of Chemistry, Idaho State University, Pocatello, ID 83209, USA; anibrose@isu.edu (R.A.); dejekarl@isu.edu (K.D.J.); pashsrin@isu.edu (S.P.); 2School of Chemical Engineering, Yeungnam University, Gyeongsan 38541, Korea; samjhanapradhan313@gmail.com; 3Biomedical and Pharmaceutical Sciences, Kasiska Division of Health Sciences, College of Pharmacy, Pocatello, ID 83209, USA; sameenamateen@isu.edu

**Keywords:** *Artemisia*, extraction, isolation, bioactive compounds, biological activity

## Abstract

Diverse medicinal plants such as those from the genus *Artemisia* have been employed globally for centuries by individuals belonging to different cultures. Universally, *Artemisia* species have been used to remedy various maladies that range from simple fevers to malaria. A survey conducted by the World Health Organization (WHO) demonstrated that 80% of the global population is highly reliant on herbal medicine for their primary healthcare. WHO recommends artemisinin-based combination therapies (ACT) for the treatment of global diseases such as malaria. Artemisinin is a bioactive compound derived from *Artemisia annua* leaves. It is a sesquiterpene endoperoxide with potent antimalarial properties. This review strives to instill natural products to chemists and others in diverse fields with a heterogeneous set of knowledge compiled from multifaceted researchers and organizations in literature. In particular, the various *Artemisia* species and effective extraction, isolation, and characterization methodologies are discussed in detail. An in-depth investigation into the literature reveals that divergent species of *Artemisia* exhibit a vast array of biological activities such as antimalarial, antitumor, and anti-inflammatory activities. There is substantial potential for bioactive compounds from *Artemisia* to provide significant relief from differing human ailments, but more meticulous research in this field is needed.

## 1. Introduction

The beauty of natural products chemistry can be witnessed in the sheer diversity of its sources. Plants, animals, and microorganisms contain a vast quantity of bioactive compounds. Plants specifically present remedies for the treatment of various anthropogenic ailments in several continents around the globe such as in Asia, Africa, and South America [1]. Traditional medicine involving plants is used by 80% of the global population and serves as the primary healthcare system [2]. The role compounds derived from natural sources play in the drug discovery process can never be overemphasized. *Artemisia* has indiscriminately served as a treatment for maladies such as viruses, malaria, bacteria, hepatitis, fungi, cancer, and inflammation [3]. Thus, the biodiversity of nature is something that is to be cherished and upheld. Among the many appealing aspects of biodiversity, the biodiversity of nature signifies numerous untapped opportunities to discover novel plant secondary metabolites. These secondary metabolites originating from flora serve the plant by protecting it against pathogens and herbivores. The primary classes of secondary metabolites for *Artemisia* species are phenolic compounds, terpenoids, and alkaloids, coumarins, acetylenes, sterols, and caffeoylquinic acids [1,2].

With the distinctive therapeutic and medicinal properties that *Artemisia* L. possesses, it would be quite remiss to not discuss its taxonomic classifications to arrive at a more holistic perspective of its scope and potential. The genus, *Artemisia* L., is considered to be one of the largest enumerable and dispersed genera in existence [2]. It is a member of the Asteraceae family (Compositae), a sizable taxonomic classification, which encompasses approximately 1000 genera and 20,000 species [3]. *Artemisia* is a member of the Anthemideae tribe, which is inclusive of over 500 species; these species are predominantly located in geographic regions such as North America, Europe, and Asia [3]. The species of *Artemisia* are characterized by their annual, biannual, and perennial herbs or compact shrubs.

A superb advantage of the *Artemisia* genus is its ability to thrive and persist in nearly all habitat types. To better comprehend the primacy of *Artemisia*, it is worthwhile to recognize its global presence. *Artemisia* is indigenous to Europe, North America (United States and Canada), South America (Brazil) Southeast Asia, South Africa, and the Pacific Islands [4]. The preferred climate for this genus is arid and semi-arid. With the exception of Antarctica, *Artemisia* is incontrovertibly well dispersed on the global arena [5]. Its far-reaching distribution allows for its unique morphology and characteristics [5]. *Artemisia* effortlessly adapts to a wide range of temperatures but thrives in moist soils [4]. The capability of this species to be able to adapt and survive in both warm and cool temperature environments provides a highly valuable protective mechanism against extinction. Field pictures of eight *Artemisia* species are present in the Figure 1.

### Traditional and Current Uses of Artemisia Species

*Artemisia absinthium* L. is typically referred to as wormwood and is a perennial plant that is dispersed and distinguishable across parts of Siberia and Europe [3]. It produces vibrant yellow flowers and possesses a wide array of uses. For instance, its antiparasitic properties allow for the treatment of anorexia and indigestion [3]. Moreover, the aerial parts of this shrub can serve as a component in various gastric herbal concoctions, intoxicating beverages, and dietary supplements [3]. In the ethnopharmacological sphere, *Artemisia absinthium* L. has functioned to restore diminished mental performance, to alleviate liver inflammation, and to enhance memory [2]. The antioxidant activity of the aerial components of *A. absinthium* has been evaluated by measuring the free-radical scavenging activity of *A. absinthium* extracts to remove reactive hydroxyl radicals and stable 2,2-diphenyl-1-picrylhydrazyl (DPPH) free radical “during the Fenton reaction trapped by 5,5-dimethyl-1-pyrroline-N-oxide”, along with the use of electron spin spectroscopy [2]. Prior research has conveyed that crude ethanol and aqueous extracts of the aerial parts of *A. absinthium* display anthelmintic activity when juxtaposed to the effectiveness of albendazole “against the gastrointestinal nematodes of sheep” [2].

*Artemisia annua* L. is commonly known as “huanghuahao” (yellow flower *Artemisia*) and it distributes from north to south to almost all parts of China [7]. It is also native to Tanzania, Kenya, India, Romania, Inner Mongolia and Hebei provinces and Vietnam [8]. In Chinese culture, *A. annua* has been utilized to address fever and chills since the second century B.C. [3,7]. This plant has been introduced to disparate parts of the world. For example, in Africa, *A. annua* is cultivated and employed in tea form to treat malaria [3]. A clandestine research project known as Project 523 was postulated in a meeting on 23 May 1967, amid the Vietnam War [7]. Evidently, the death toll experienced by North Vietnamese soldiers during battle was eclipsed by the number of soldiers that met their premature demise by malaria. Thus, the search began to discover antimalarial compounds for both the Chinese citizens and the North Vietnamese soldiers. In 1972, a year that marked a turning point for the health of a nation, artemisinin was extracted and recognized for its highly efficacious anti-malarial activity from *A. annua* [7]. Initially, the potency of this anti-malarial compound, artemisinin, was trivialized by the Western world [7]. Nonetheless, artemisinin repeatedly demonstrated its effectiveness against schistosomiasis and malaria [7]. In addition to *A. annua’s* antimalarial efficacy, it has been proposed that *A. annua* can also effectively combat human immunodeficiency virus (HIV) [9]. Therefore, more recent research investigations are involved in determining its antiviral activity against HIV [9]. This is especially pertinent because HIV is a prolific global disease.

*Artemisia biennis* is typically known as biennial wormwood and is an annual or biennial species naturally present in western North America [10]. The spreading of *A. biennis* throughout North America occurred as a result of anthropogenic activities such as transportation [10]. The proliferation of *A. biennis* in agricultural lands can be ascribed to a forbearance of some classes of herbicides, shift in annual growth patterns, crop diversification, and a rise in reduced tillage systems usage [10,11]. *A. biennis* thrives in highly disturbed habitats and can effortlessly outperform species native to a certain habitat. In this fashion, biennis is considered an invasive species. Although this herb can be quite a quandary in certain regions, it does possess some beneficial uses. This species commonly functions as an antiseptic in folk remedies and also used for cooking purposes [9]. Furthermore, *A. biennis* has been employed by the indigenous people of North America as ointments and medicinal cleansers for the treatment of wounds, sores, and chest infections [3]. In Iran, the essential oil of this herb is primarily composed of camphor [12]. However, in Western Canada, the most copious volatile compound in the aerial portions of *A. biennis* is €-β-farnesene [8,12]. Mojarrab et al. (2016) carried out an investigation that sought to determine the efficacy of fractions derived from *A. biennis* hydroethanolic extract to grant “cytoprotection against oxidative stress and apoptosis induced by doxorubicin (DOX) in rat pheochromocytoma cell line (PC12)” [8,12]. This investigation corroborated that oxidative stress injury as well as “apoptosis induced by DOX in the PC12 cells” was attenuated by the hydroethanolic extract of *A. biennis* [8,12].

*Artemisia campestris* L. is frequently referred to as tgouft and is a faintly fragrant perennial herb widely distributed in the south of Tunisia, North Africa, North America, and Eurasia [3,10,13]. The leaves and flowers of *A. campestris* were conventionally implemented for the treatment of high cholesterol levels, obesity, hypoglycemia, and venin [13]. Innumerable studies attest to *A. campestris’s* bioactivity. The biological activities of this plant include allelopathic, anthelmintic, antimicrobial, antidiabetic, hepatoprotective, insecticidal, nephroprotective, antivenomous, antiulcer, antioxidant, and antitumor [13]. The study conducted by Ivanescu et al. (2018) on *Artemisia campestris* from Romania divulged novel bioactive compounds such as acacetin, casticin, eupatorin, stigmasterol, α-sitosterol, gentisic acid, and campesterol through various biological assays and LC-MS [10,13].

*Artemisia douglasiana* L. is commonly referred to as California mugwort and is a perennial herb native to the Western United States, particularly in northern California, Washington, and Oregon [9]. In Cuyo, a region in which *A. douglasiana* is nonnative, it is cultivated and used in traditional medicine [3]. The natives of Cuyo refer to *A. douglasiana* as “matico”; it is employed for the treatment of gastrointestinal disorders and peptic ulcers [3]. Furthermore, *Artemisia douglasiana* serves as a tonic to remedy nervous disorders and to stimulate menstruation [9]. The essential oil of *A. douglasiana* has been utilized for aromatherapy, relieving tense muscles, mollifying mental anguish, and inhalation to enhance mental clarity [9]. According to Setzer et al. (2004), the leaf oil from *A. douglasiana* analyzed by GC-MS revealed the prime components of the oil to be 29% camphor, 26% *Artemisia* ketone, 13% *Artemisia* alcohol, 10% alpha-thujone, 8% 1,8-cineole, and 15% hexanal [12,14].

*Artemisia dracunculus* L. is also known as “tarragon” and is considered a perennial herb widely distributed in western North America, eastern and central Europe, and a majority of temperate Asia [15]. *A. dracunculus* is also cultivated in India, Russia, Iran, and Ukraine [16]. This herb possesses an extensive background in culinary practices. In America, tarragon is used in seafood, chicken, eggs, tartar sauce, and vinegar [15]. In France, tarragon is used to flavor French Dijon mustard, sour cream, eggs, and mayonnaise [15]. Due to its diverse and expansive health promoting properties, it typically functions as herbal medicine [3]. For instance, extracts of *A. dracunculus* have been made use of by Himalayan natives to pacify toothache, relieve fever, and to remedy gastrointestinal issues [9]. Tarragon possesses phenomenal antibiotic activity; in Tibetan medicine, this plant is used to treat chronic bronchitis, pulmonary tuberculosis, and pneumonia [16]. Native American Chippewa tribe utilized the roots of tarragon to mitigate superfluous flow during the menstruation period and to assist in strenuous labor [15]. The dominant essential oil components of *A. dracunculus* consist of coumarins, flavonoids, and phenolic acids [15].

*Artemisia tridentata* Nutt. is informally known as “basin big sagebrush” and is a perennial shrub that possesses a vast range of distribution [9]. This shrub is native to the semiarid Southwestern United States [17]. *Artemisia tridentata* encompasses the following subspecies: *tridentata, vaseyana*, *wyomingensis*, *rupicola*, *xericansis*, *scopulorum*, and *thermopola*. *A. tridentata* serves as a consequential habitat for a great variety of fauna [9,14]. Big sagebrush boasts an assortment of uses. Native Americans utilize this shrub to remedy flu and colds [17]. Historically, *A. tridentata* has been employed for the alleviation of fevers, poisoning, and gastrointestinal issues. Although basin big sagebrush is frequently lauded for its appreciable chemotherapeutic and bactericidal activities, it is an efficacious aeroallergen and liable to cause dermatitis [17]. The most salient compounds present in *A. tridentata* include sesquiterpenes, flavonoids, monoterpenes, and coumarins [17].

*Artemisia vulgaris* L. is quite a significant medicinal plant species that contains volatile oils. This plant is known by many endearing and unique names, but perhaps the most common being Mugwort [1]. Interest in *Artemisia* is specifically related to its phenomenal biological and chemical biodiversity along with the discovery of antimalarial drug artemisinin. *Artemisia* vulgaris has a prominent history distinguished by its use as a medicinal plant for the alleviation of human afflictions globally [1]. *Artemisia* vulgaris possesses a diverse array of medicinal properties inclusive of anti-septic, anti-inflammatory, anti-spasmodic, anti-hypertensive, hepatoprotective, anti-oxidant, and anti-tumoral [1]. The secondary metabolites in this plant are responsible for these valuable properties. Some of the more prominent classes of secondary metabolites are flavonoids, sesquiterpene lactones, phenolic acids, acetylenes, and coumarins [1]. Pharmacological and phytochemical investigations have confirmed the medicinal capability of bioactive compounds [1]. Due to the versatility of *A. vulgaris*, demand continues to soar in an unsustainable fashion.

A great variety of flavonoids has been isolated from *Artemisia* vulgaris; the primary classes are flavones, flavone glycosides, flavanols, and flavonol glycosides [5]. In the case of the flavones class: tricine, apigenin, eupafolin, luteolin, diosmetin, jaceosidine, and chrysoeriol were extracted. For flavone glycosides, vitexin and luteolin 7-glucoside were extracted. Flavones eluted include eriodictyol and homoeriodictyol. As for the flavonols class, only isorhamnetin was isolated. The compounds within the class of flavonol glycosides that were extracted includes kaempferol 3-glucoside, rutin, kaempferol 3-rhamnoside, quercitrin, kaempferol 3-rutinoside, and quercetin 3-galactoside [5]. Luteolin and eriodictyol were determined to be the most prominent compounds isolated from vulgaris [18].

Phytonutrients such as fruits and vegetables contain fantastic pharmacological properties; pharmacological properties such as anti-inflammatory, antioxidant, and anti-carcinogenic reside within certain fruits and vegetables [19]. For instance, Eriodictyol, a flavanone, is present in citrus fruits. Eriodictyol possesses anti-inflammatory properties which can garner tremendous aid for the care of diabetic retinopathy [20]. Luteolin is commonplace in the food industry and is notably utilized in oregano, peppermint, olive oil, and plant-based food. Luteolin is a bioactive compound replete with a wonderful array of biological properties such as antioxidant, anticancer, antimicrobial, and anti-inflammatory (in vivo and in vitro) [21].

*A. vulgaris* is not a significant source of artemisinin when compared to its presence in other *Artemisia* species such as *Artemisia* annua [22]. Essential oils of *A. vulgaris* comprise α-thujone, α-pinen, β-caryophyllene, camphor, 1,8-cineole, camphene, and germacrene [5]. The essential oils that were isolated appeared to assume a green hue in conjunction with a pervasive camphoraceous scent and a bitter-sweet taste. These characteristics can be imputed to the presence of an appreciable quantity of oxygenated monoterpenes and sesquiterpenes in the oil of the stem. The oxygenated monoterpene and sesquiterpene content present in the extracted essential oils of *A. vulgaris* have been evaluated: monoterpenes (camphor [17.3%], alpha-thujone [10.7%], and 1,8-cineole [5.1%]) and sesquiterpene derivatives (gamma-muurolene [9,0%] and beta-caryophyllene [5.8%]) [23].

The presence of essential oils in *A. vulgaris* are reliant on aspects such as environmental conditions, geographic derivation, and stages of plant development [5]. Moreover, the distinct oils contained in specific plant parts were monitored in varied biological stages to gauge other facets that played noteworthy roles in the quantity and quality of the essential oils. Unsurprisingly, stress factors and population genetics also contribute to the quantitative and qualitative compositions of *A. vulgaris* in dissimilar regions [5]. As evidenced by the GCMS analysis, essential oils in Mugwort from Morocco were found to be far more profuse in camphor and isothujone/thujone ratio when compared to Mugwort from Iran which was abundant in α and β-pinene [24,25]. Furthermore, in comparing essential oils from Italy and India, it has been reported that Mugwort from Italy is significantly composed of camphor (47%) or camphor (2–20%) with myrcene (9–70%), while vulgaris from India is principally composed of camphor (38.7%) and isoborneol (8.2%) [26,27].

According to the literature, camphor is one of the most prevalent essential oils in *A. vulgaris*, even in dissimilar regions globally. The increase in camphor concentration has been determined to accumulate in the following order: seeding stage, flowering, buds, and vegetative [5]. In 2006, the disparities in the chemical composition of the essential oils in the root and aerial parts of *A. vulgaris* were confirmed to demystify the influence plant parts and storage had on the composition and yield of essential oils. The essential oils extracted from the aerial parts of the plant divulged a copious amount of 1,8-cineole (28.9%), sabinene (13.7%), βthujone (13.5%), and β-caryophyllene oxide (6.5%) [28]. Conversely, the oils of primacy isolated from the root were neryl 2-methylbutanoate (13.2%), β-eudesmol (10.0%), and bornyl 3-methylbutanoate (8.4%) [28]. This indicates that the location in which essential oils are extracted from a plant is consequential.

The utility of *A. vulgaris* in various geographic locations and cultures appears endless. Mugwort has been known to possess an array of therapeutic properties and applications; it has conventionally been employed to remedy indigestion, liver disorders, and stomach ulcers [29]. Furthermore, it has been reported that the entire plant can alleviate diabetes, insomnia and stress, depression, anxiety, epilepsy, and worm infestation. It can also be used to stimulate delayed or inconsistent menstruation [29]. The leaves and stem of Mugwort have served as a uterine stimulant and digestive tonic. Alternatively, to season fish and meat, the leaves and buds of Mugwort can be quite useful in preparing a rather savory meal. For instance, in Asia, this herb has reportedly been utilized to add flavor to tea and traditional cuisines [26].

The antioxidant capability of *A. vulgaris* has been explored by Sharmila and Padma (2014) using methanol extract in chick embryos exposed to oxidative stress [30]. This demonstrated that in the presence of *A. vulgaris*, the antioxidant activity of primary cells that were placed under oxidative stress were enhanced [30]. In another study carried out by Haniya and Padma (2014), when exposed to the free radical agents such as DPPH, ABTS, hydrogen peroxide, superoxide, hydroxyl radicals, and nitric oxide, the solvent extract that demonstrated the optimal scavenging effects was the methanol extract [31]. This corroborated the antioxidant strength of *A. vulgaris* leaf extract.

In an experiment conducted by Afsar et al. (2013), the anti-inflammatory activity of Mugwort was analyzed with the cotton pellet granuloma method. Once cotton pellets were surgically implanted into the groin of rats, methanol leaf extracts were delivered at doses of 200 and 400 mg/Kg body weight [5,32]. It was determined that at 400 mg/Kg, there was a 64.0% inhibition in weight of dry cotton pellets and 55.3% inhibition in weight of wet cotton pellets in comparison to the control [5,32]. This dose-dependent anti-inflammatory characteristic can be attributed to flavonoids present in *A. vulgaris*.

## 2. Extraction, Isolation, Characterization

The aim of this review is to address the process involved in order to obtain pure compounds from crude plant extracts, which includes the combination of extraction/sample preparation tools and analytical techniques, for isolating and characterizing bioactive compounds from plants, as potential lead compounds in the drug discovery process. A flowchart of stages involved in extraction, isolation and characterization of bioactive compounds from *Artemisia* is present in Figure 2.

### 2.1. Extraction Techniques

Fauna and flora possess an extensive array of bioactive compounds that can be applied to the enhancement of human wellbeing. However, the extraction of these bioactive compounds by various research groups can be carried out in a more sustainable fashion that can preserve the abundance and richness of the species used in research. Conventional extraction methodologies are entrenched in the employment of vast quantities of organic solvents only to result in less than satisfying extraction yields [33]. Thus, it is imperative and worthwhile for green extraction techniques to be explored. Such techniques prioritize the avoidance of toxic solvents and the increase of extraction yields with minimal energy expenditure. Currently, supercritical fluid extraction (SFE) and pressurized liquid extraction (PLE) are ubiquitous high-pressure techniques for the elution of bioactive compounds from natural sources [33]. These techniques possess considerably high extraction efficiencies [33]. Some of the extraction technique used for the extraction of bioactive compound of *Artemisia* species has been mentioned in Table 1.

Supercritical fluid extraction (SFE) makes use of a specific solvent under temperatures and pressures beyond its critical point. Under such conditions, the fluid undergoes “physicochemical changes” that impart alterations to its solvent properties [33]. Consequently, no phase separation is observed, and a homogenous supercritical fluid is obtained. This enables supercritical fluids to achieve viscosities akin to gaseous substances while simultaneously retaining densities homologous to liquid substances. These characteristics, in conjunction with others, are what distinguish supercritical fluids from more orthodox solvents found under ambient parameters.

The SFE technique can be utilized for the extraction of bioactives from a great variety of natural resources [33]. Carbon dioxide is perhaps the most prominent supercritical fluid made use of for bioactive extraction [33]. The ubiquitous use of supercritical CO_2_ (sc-CO_2_) can be ascribed to its affordability, safeness, moderate critical temperature and pressure (31.2 °C and 73.8 bar, respectively) that is effortlessly attainable, and FDA designation of Generally Recognized As Safe (GRAS) for the food industry [33]. Moreover, CO_2_ is considered to be a by-product of various industrial processes [33]. Therefore, the use of CO_2_ is considered environmentally friendly since it is being reused. Another compelling facet of CO_2_ that makes it an appealing solvent alternative is that its gaseous phase can be maintained at room temperature [33]. This allows for direct and effortless evaporation of CO_2_ from the extract upon the completion of the extraction procedure [33]. Temperature and pressure play an appreciable role in the extraction process since their unique combinations inherently determine the density of sc-CO_2_ and thereby the capability to selectively elute certain substances from the extract matrix [33]. As with most systems, there does exist a caveat for this technique. The polarity of sc-CO_2_ is fairly low, which can be interpreted as polar bioactive compounds being neglected in the extraction process [33]. However, with the use of a modifier or a co-solvent such as ethanol in small quantities, the polarity of sc-CO_2_ can be enhanced [33]. Thus, expanding the range of bioactive compounds that can be extracted. 

The other illustrious high-pressure green technique is called pressurized liquid extraction (PLE). Pressurized liquid extraction is also referred to as accelerated solvent extraction (ASE) or pressurized fluid extraction (PFE) [33]. PLE is a technique characterized by the employment of suitable pressures to sustain the solvent in its liquid phase and high temperatures that lay below the critical point [33]. When juxtaposed to more conventional methods, the merits of PLE can clearly be appreciated. For instance, a smaller amount of solvent is needed to carry out the technique. As a result, the efficiency of extraction is enhanced due to a better mass transfer rate. Under elevated temperature conditions, solvent solubility is improved while viscosity decreases; this aids in matrix penetration and mass transfer [33]. Doubtlessly, pressure and temperature are consequential parameters to be considered when performing PLE. However, while pressure is a significant parameter, increasing pressure beyond the point necessary to maintain the solvent in its liquid state has a negligible impact on the extraction process [33]. Instead, time would be better spent on selecting a solvent that is consistent with the nature of the desired compounds.

The considerably low concentrations of bioactive compounds in *Artemisia* species presents a profound challenge that can sometimes derail the progress of their analysis. The extraction of bioactive compounds free of interferents using conventional methods is nearly unattainable. Hydrophilic deep eutectic solvents (DESs) have captured the attention of many and are considered green alternatives for the efficient extraction of bioactive compounds [34]. In an investigation carried out by Jun Cao et al. (2017), the merits of DESs are illuminated and compelling incentives for the use of DESs are discussed. A considerable amount of DESs is typically composed of two or more non-noxious, non-flammable, biodegradable, and inexpensive components that interact with one another via hydrogen bonding [34]. Moreover, deep eutectic solvents possess a wide range of hydrogen bond donors (HBDs) and hydrogen bond acceptors (HBAs) which make them applicable in a wide array of fields due to their minimal toxicity, simple preparation, biodegradability, and unique properties [34].

The fundamental intent of this study was to evaluate whether DESs could improve the concentration of artemisinin extracted from *Artemisia* annua, thereby improving the extraction efficiency. To achieve this end, a series of “screening and tailoring” was carried out to determine which DES would produce the greatest extraction yield^30^. Subsequently, a hydrophobic DES known as N81Cl-NBA was chosen as the solvent for extraction due to its high extraction yield. N81Cl-NBA was prepared from methyl trioctyl ammonium chloride and 1-butanol using a molar ratio of 1:4 [34]. N81Cl-NBA-based ultrasound-assisted extraction was employed, and the components influencing extraction yield were statistically optimized [34]. Jun Cao et al. (2017) concluded that N81Cl-NBA-based ultrasound-assisted extraction along with “macroporous resin separation” demonstrated a higher extraction efficiency and an artemisinin recovery yield of 85.65% was achieved [34]. Compared to 60–80% yield of artemisinin using conventional organic solvents such as petroleum ether, this is a pleasing improvement [34]. This investigation was able to demonstrate that DESs are “designer solvents” that can be utilized as green extraction solvents for the extraction of bioactive compounds from plant material [34].

**Table 1 molecules-26-06995-t001:** Bioactive compound extraction techniques for *Artemisia* species.

Species Name	Origin	Method	Plant Part	Solvent System	Bioactive Constituent	Percent Yield (%)	Reference
*Artemisia annua*	Astore, Northern Areas Pakistan	Sonication	Flowers	5 mL HPLC grade toluene	Artemisinin	0.42 ± 0.03%	[35]
*Artemisia annua*	Astore, Northern Areas Pakistan	Sonication	Leaves	5 mL HPLC grade toluene	Artemisinin	0.44 ± 0.03%	[35]
*Artemisia annua*	Astore, Northern Area Pakistan	Sonication	Stems	5 mL HPLC grade toluene	Artemisinin	0.8 ± 0%	[35]
*Artemisia dracunculus var dracunculus*	Abbass Pur, Azad Kashmir Pakistan	Sonication	Leaves	5 mL HPLC grade toluene	Artemisinin	0.27 ± 0%	[35]
*Artemisia dracunculus* *var dracunculus*	Abbass Pur, Azad Kashmir Pakistan	Sonication	Stems	5 mL HPLC grade toluene	Artemisinin	0.12 ± 0.01%	[35]
*Artemisia parviflora*	Rawalakot, Azad Kashmir Pakistan	Sonication	Stems	5 mL HPLC grade toluene	Artemisinin	0.8 ± 0%	[35]
*Artemisia moorcroftiana*	Kalam, Swat Pakistan	Sonication	Stems	5 mL HPLC grade toluene	Artemisinin	0.8 ± 0%	[35]
*Artemisia sieversiana*	Soost, Northern Areas Pakistan	Sonication	Roots	5 mL HPLC grade toluene	Artemisinin	0.04 ± 0%	[35]
*Artemisia sieversiana*	Soost, Northern Areas Pakistan	Sonication	Stems	5 mL HPLC grade toluene	Artemisinin	0.8 ± 0%	[35]
*Artemisia moorcroftiana*	Kalam, Swat Pakistan	Sonication	Stems	5 mL HPLC grade toluene	Artemisinin	0.8 ± 0%	[35]
*Artemisia vestita*	Galyat, Pakistan	Sonication	Roots	5 mL HPLC grade toluene	Artemisinin	0.04 ± 0%	[35]
*Artemisia vulgaris*	Kalam, Swat Pakistan	Sonication	Flowers/leaves	5 mL HPLC grade toluene	Artemisinin	0.05–0.15%	[35]
*Artemisia vulgaris*	_	silica gel column chromatography using gradient elution	Leaves	Ethyl acetate and dichloromethane	Yomogin and 1,2,3,4-diepoxy-11(13) eudesmen-12,8-olide	_	[36]
*Artemisia douglassiana*	_	Liquid-liquid extraction	Aerial parts	Ethyl acetate-hexane [1:9]	Dehydroleucodine	_	[36]
*Artemisia douglassiana*	_	Liquid-liquid extraction	Aerial parts	Silica gel with hexane-ethyl acetate mixtures	Dehydroparishin-B	_	[36]
*Artemisia diffusa*	_	Maceration	_	n-hexane/ethyl acetate/methanol [1:1:1]	Tehranolide	_	[36]
*Artemisia* *princeps*	_	Liquid-liquid extraction	Aerial parts	Dichloromethane fraction column chromatography over silica gel using gradient elution of methanol and dichloromethane	Yomogin	_	[36]
*Artemisia ludoviciana*	_	Column chromatography	Aerial parts	Organic phase chromatographed repeatedly on normal-phase silica gel with ethyl acetate and hexane	Guaianolide ludartin	_	[36]
*Artemisia* *caerulescens* ssp. *cretacea*	_	Concentrated extract is extracted with boiling water. Aqueous solutions are extracted with chloroform	Flowers	Aluminum oxide column with 5% methanol in chloroform	Santonin	_	[36]

### 2.2. Isolation Techniques: TL

Typically, plant extracts are composed of a mixture of bioactive constituents that need to be isolated for the characterization process [37]. Thin layer chromatography (TLC) is a commonplace technique for the separation of mixtures [37]. TLC analysis is a straight-forward, efficient, and inexpensive method that provides the researcher with immediate results on the quantity of the components present in the mixture [38]. The performance of a TLC analysis can also be employed to corroborate the identity of a compound in a mixture by comparing the retention factor (R_f_) of the compound in the mixture with a known compound [38]. Subsequently, the components of the mixture on the TLC plate can be visualized with the use of a UV lamp [37].

In a study conducted by Mohammad Bagher Pasha Zanousi (2012), *Agrobacterium rhizogenes* promoted the formation of hairy roots in an “Iranian clone of *Artemisia annua*” to evaluate the production of artemisinin in the newly formed hairy roots by TLC analysis [38]. To begin the TLC analysis, the artemisinin standard was dissolved in HPLC grade acetonitrile to produce a 1000 ppm concentration of artemisinin [38]. The anhydrous extracts were placed in 1 mL of acetonitrile. Additionally, a micropipette was used to spot the silica gel TLC plate. The TLC plate was subsequently placed in a glass TLC chamber containing a solvent system of acetone/hexane with a 3:10 molar ratio [38]. The plate was developed to a height of 3 cm and left to air dry at room temperature [38]. The dry TLC plate was visualized in an iodine tank [38]. Finally, artemisinin was identified by comparing the color intensity of the artemisinin standard to the other extracts [38].

The TLC analysis provided insight into the concentration of artemisinin in the standard and various root types evaluated. The artemisinin standard (1000 ppm) contained the highest artemisinin concentration since it was the darkest spot on the TLC plate. In comparing the color intensity of the hairy roots extract and the control roots extract on the TLC plate, it was determined that the hairy roots extract possessed a higher concentration of artemisinin than the control roots. In essence, this research confirmed that inducing the formation of hairy roots in the leaves of *A. annua* proved effective for the increased production of artemisinin.

High-performance liquid chromatography (HPLC) is a ubiquitous technique utilized for the isolation of secondary metabolites. Natural products such as secondary metabolites are typically isolated after the examination of the crude extract in a biological assay; this is performed to achieve a more holistic characterization of the bioactive constituents [37]. The meager quantities of bioactive compounds in the extract make the resolving power of HPLC critical for the swift processing of multi-component plant samples on a preparative and analytical scale [37]. Currently, numerous benchtop HPLC instruments possess a modular design and are inclusive of a solvent delivery pump, an auto-sampler or manual injection valve, a guard column, an analytical column, a detector, and a recorder or a printer [37]. HPLC can be employed for chemical separations because the constituents of the extract possess distinct migration rates and certain parameters are established. Ultimately, the mobile phase and stationary phase chosen determines the extent of separation [37]. An isocratic system (makes use of a single mobile phase system) is the general method utilized for the separation and identification of phytochemicals. However, if multiple sample components are of interest to the researcher, it would be ideal to employ gradient elution that involves the modification of the organic and aqueous proportions over a specified amount of time [37]. Gradient elution may prove challenging if the analytes possess analogous properties and similar retention times.

In HPLC, purification can be achieved by separating the compound of interest from other compounds or interferents. Based on the chromatographic conditions selected, each compound will possess a characteristic peak. To separate a variety of compounds, the chromatographer has the unique ability to tamper with parameters such as the mobile phase, column type, flow rate, and detectors to ascertain optimal peak resolution and separation [37]. For the identification of compounds, a suitable detector for HPLC must be chosen [37]. The chosen detector should be set to appropriate detection settings and a method is to be developed to yield a clean peak on the chromatogram for the specific analyte being analyzed [37]. UV detectors are quite commonplace among researchers because they afford high sensitivity [38]. Furthermore, phytochemicals of research interest commonly have a UV absorbance at low wavelengths that ranges from 190–210 nm [37]. Apart from UV detectors, other detectors such as the diode array detector (DAD) coupled with a mass spectrometer are employed to assess phytochemicals.

There have been several studies that have been successful in quantifying bioactive compounds present in plant material. The paramount purpose of the research performed by Sakipova et al. (2017) was to establish an efficient and distinct method for santonin identification and quantification with the use of HPLC [39]. The researchers more specifically desired to diminish unwarranted intoxication of santonin present in *Artemisia cina* so that its anthelmintic properties can be used in veterinary medicine [39]. The method developed from this research was utilized to characterize and quantify santonin levels in the leaves of eight distinct *Artemisia* species: *A. cina*, *A. scoparia*, *A. absinthium*, *A. terra-albae*, *A. gmelinni*, *A. sublesingiana*, *A. schrenkiana*, and *A. frigida* [39]. The primary result of this investigation demonstrated that santonin in *Artemisia* cina possessed a retention time was approximately 5.7 min, a time congruent with the Santonin standard tested [39].

The HPLC system (Waters) was employed for the quantitative analysis of santonin equipped with the breeze software program, Waters 717 plus autosampler, Waters 1525 binary HPLC pump, and Waters 2487 dual wavelength absorbance detector [39]. The solvent system applied was water (solvent A) and acetonitrile (solvent B). The gradient elution program utilized was: 35% A—65% B at 0 min, 35% A—65% B for 5 min, 45% A—55% B for 10 min, 55% A—45% B for 15 min, and 65% A—45% B for 20 min [39]. The chosen HPLC system parameters for the wavelength and pressure were 236 nm and 5000 atm, respectively [39]. HPLC chromatogram for chloroform extract from *Artemisia* cina is shown in Figure 3. 

In another investigation led by Tian et al. (2020), six phenolic acids derived from *Artemisia capillaris* (Yinchen) were analyzed using HPLC and their transformation pathways assessed during the decoction process [40]. The intent of this research was to establish and corroborate a novel analytical protocol for the analysis of phenolic acids such as 4,5-dicaffeoylquinic acid, 3,4-dicaffeoylquinic acid, 1,3-dicaffeoylquinic acid, 4-caffeoylquinic acid, and 3-caffeoylquinic acid to achieve “quality control” of *A. capillaris* decoction [40]. To accomplish this purpose, HPLC coupled with diode array detection (HPLC-DAD) was employed [40].

The HPLC system utilized for the quantitative investigation of the six phenolic acids was a Thermo UltiMate 3000 (USA) containing an autosampler, a column temperature controller, and a DAD (190–300 nm) [40] is shown in Figure 4. To maintain the column temperature at 30 °C, an Agilent Eclipse XDB-C_18_ column (250 mm × 4.6 mm) with a particle size of 5 µm was used [40]. The applied solvent system was a 0.1% formic acid in water (solvent A) and acetonitrile (solvent B) [40]. The gradient elution program employed was: 7–15% B for 0–20 min (linear gradient), 15–20% B for 20–30 min (linear gradient), 20% B for 30–35 min (isocratic gradient), and 20–25% for 35–45 min (linear gradient) [40]. With an injection volume of 5 µL and a 5 min post-run at a flow rate of 1.0 mL/min, a high resolution of the desired compound peaks was achieved [40].

### 2.3. Characterization: LCMS & NMR

Liquid chromatography coupled with a mass spectrometer (LC-MS) is a salient and efficacious technique utilized for the analysis of intricate botanical extracts [37]. With the simultaneous application of tandem mass spectrometry, LC-MS can provide ample data for the structural determination of bioactive compounds [37]. The research carried out by Wang et al. aimed to develop an LC-MS method accompanied by selected ion monitoring (SIM) for the quantification of artemisinin from various locations [41]. To track “the abundance of the [M-18+H]^+^ ion peak at *m*/*z* 265.5”, SMI was employed, and a scan range from *m*/*z* 250 to 270 was used [41]. As a result of monitoring the [M-18+H]^+^ ion peak, LC-MS sensitivity was propitiously enhanced [41] as shown in Figure 5. The method developed from this research offers a straightforward and accurate manner to detect and quantify artemisinin with a total run time of 11 min per sample [41].

The motive behind the research conducted by Pratibha Singh et al. in 2020 was to identify and determine bioactive compounds in the ethanolic extracts of various *Artemisia* species for anti-malarial and anti-diabetic purposes shown in Figure 6 [42]. To achieve this end, the researchers established the “Ultra Performance Liquid Chromatography coupled with hybrid triple quadrupole linear ion trap tandem mass spectrometry (UPLC-ESI-QqQ_LIT_-MS/MS) method in multiple reaction monitoring (MRM) acquisition mode” [43]. The method developed was then substantiated using recovery, linearity, stability, precision, limit of detection (LOD), and limit of quantitation (LOQ) [42]. There are several analytical techniques in existence that have been applied to probe the nature of a variety of bioactive constituents innate in *Artemisia* species such as gas chromatography–mass spectrometry (GC-MS), high-performance liquid chromatography (HPLC), and high-performance thin-layer chromatography (HPTLC) [42]. However, the previously mentioned techniques are considered to be lacking due to a variety of limitations, some of which may be attributed to their lower selectivity and sensitivity [43]. With a more advanced technique such as the UPLC-ESI-QqQ_LIT_-MS/MS), which is an accurate and largely accepted technique, copious metabolites in complex plant extracts can be identified and quantified by applying multiple reactions monitoring (MRM) mode [42]. In this study, the developed method was used to investigate the bioactive components in different *Artemisia* species such as *A. vulgaris*, *A. absinthium*, *A. dracunculus*, *A. annua*, *A. maritima*, *A. vestita*, and *A. verlotiorum* [42].

Nuclear magnetic resonance (NMR) spectroscopy is a remarkably useful instrument for the investigation of the chemical constituents in extracts of a specific sample. Researchers have utilized NMR extensively in search of drug candidates derived from natural products, structure-based drug design, and drug discovery [43]. In a study performed by Nageeb et al. (2013), ^1^H NMR was employed to juxtapose the diverse metabolites contained in *Artemisia annua* garnered from Egypt and Jericho presented in Figure 7 [43]. Various organic extracts and aqueous extracts used in traditional medicine were utilized to compare the biological activities of *Artemisia annua* from Egypt (EA) and *Artemisia annua* from Jericho (JA) [43]. NRM analysis served to corroborate the differences in the bioactivity of EA and JA. The higher temperatures and arid weather conditions in Jericho afforded the JA extracts (water and methanol) greater antibacterial and antioxidant activities than the extracts from Egypt [43]. Conversely, the methanol extract of *A. annua* from Egypt demonstrated high anticancer activity while the EA water extract and both JA extracts displayed meager anticancer activity [43]. Overall, the results of this study conclude that environmental influences play an appreciable role in the various bioactive compounds present in *A. annua* species from Egypt and Jericho [43].

## 3. Bioactive Compounds from *Artemisia*

### 3.1. Phenolics & Flavonoids

As mentioned previously, a vast assortment of secondary metabolites dwells in plants. Phenolics are compounds synthesized by flora as a part of their natural development process and during stressful conditions such as UV radiation exposure, injury, and predation [44]. As a class, phenolics are structurally distinguished by their aromaticity and having a minimum of one hydroxyl group directly connected to a benzene ring [44]. Plants are a significant source of the structurally diverse class of phenolics, with over 8000 phenolics extracted from plants [44]. The structural diversity of phenolics is complemented with their protective role against eclectic ailments such as stroke, coronary heart disease, and specific types of cancer [44].

The three fundamental classes of phenolics are polyphenols (flavonoids and tannins), simple phenols (phenolic acids), terpene and a miscellaneous class [44], and they are represented in Figure 8. Polyphenols are bioactive compounds that are copious and well distributed in a great variety of plants in disparate locations [45]. Polyphenols possess antioxidant properties that serve to mollify the insidious impacts of free radicals. Polyphenols also impart color to vegetables, fruits, and flowers [45]. Typically, polyphenols are further organized into two rudimentary classes: flavonoids and tannins [44,45]. Tannins are considered to be acerbic “plant polyphenols that bind or precipitate proteins” [44]. These acerbic compounds are further organized into two primary subclasses, condensed and hydrolysable. Flavonoids are a highly diverse class of phenolic compounds. These compounds are commonly found in flowers, vegetables, stems, roots, barks, grains, fruits, tea, and wine [46]. Flavonoids are considered to be the most substantial class of phytonutrients [46]. The colors of vegetables, fruits, and other vibrantly colored parts of plants can be ascribed to the presence of flavonoids [46]. This wide-ranging class of compounds satisfy an assortment of physiological functions in flora. In the flavonoid class, there exist subclasses which are inclusive of isoflavones, flavones, flavanols, flavonols, flavanones, dihydroflavonols, bioflavonoids, chalcones, and aurones [44,45]. Various polyphenols compounds found in A. herba alba [17] and A. vulgaris [5] is shown in Figure 8 and Figure 9.

### 3.2. Terpenoids

Terpenoids are yet another consequential class of bioactive compounds present in *Artemisia* species. Terpenoids are the most abundant class of natural products in plants with over 40,000 unique structures [47]. Five-carbon isoprene units serve as the fundamental building blocks of this class. The amount of isoprene units incorporated into the structure determines the various classifications: hemiterpenes (C_5_), monoterpenes (C_10_), sesquiterpenes (C_15_), diterpenes (C_20_), triterpenes (C_30_), tetraterpenes (C_40_), and polyterpenes [47].

### 3.3. Coumarins

Naturally occurring coumarins can exist as aglycones or as glycosides and are widely distributed in more than 30 families, including about 150 species of plants. Coumarins are a class of compounds with a characteristic fragrant odor. Coumarin is the lactone derived from cis-ortho-hydroxycinnamic acid, known as benzo-α-pyrone or benzo-1,2-pyrone. They are formed in the leaves and accumulate especially in the roots and bark, as well as in old or damaged tissues. Coumarins protect plants from herbivores and pathogenic microorganisms. The main coumarins found in *Artemisia* are shown in Figure 10.

## 4. Biological Properties & Significance

The richness in the biological properties of *Artemisia* species can be imputed to the diversity of the secondary metabolites present in sagebrush [1]. Some of the most notable biological activities of *Artemisia* species are antitumor, anti-inflammatory, antiulcer, and antimicrobial [1]. Table 2 shows the health bolstering effects of *Artemisia* extracts. The biological activities of this species have considerably aided in promoting the health of millions globally [1].

### 4.1. Antitumor Activity

Artemisinin and its derivatives from *A. annua* (artemisone, artesunate, dihydroartemisinin, and artemether) are highly potent antimalarial drugs [36]. They are efficacious anticancer compounds because they are exceptionally selective to cancer cells while producing nearly minimal adverse effects on healthy cells [36]. The application of these bioactives is far-reaching and encompasses a broad spectrum of utility in leukemia, breast, melanoma, prostate, and lung cancer cells [36]. The basis of artemisinin’s antitumor mechanism is the cleavage of its endoperoxide bridge by the iron in cancerous cells along with the formation of free radicals [36]. Other bioactive compounds such as flavonoids also play a part in the holistic activity of *A. annua* extracts [45]. Specifically, the flavonoids in *A. annua* perform “synergically” with artemisinin to diminish cancer and malaria by altering the metabolism and absorption of artemisinin in the body [45].

### 4.2. Anti-Inflammatory & Immunomodulatory Activity

The sesquiterpene lactones from the *Artemisia* genus also demonstrate anti-inflammatory and immunomodulatory influence that could refine the treatment of chronic diseases and thereby, the success of therapy. The rudimentary mechanism of action for anti-inflammatory activity is the inhibition of nuclear factor κB (NF-κB). NF-κB is family of inducible transcription factors and a well know complex protein that functions to modulate 150+ inflammatory genes, DNA transcription and mediates the immune response in humans [36]. Thus, the inhibition of NF-κB minimizes inflammatory response and derails cancer proliferation. NF-κB involved in cellular responses to stimuli and has been linked to cancer, inflammatory and autoimmune diseases, septic shock, viral infection, and improper immune development [50]. In a dose-dependent fashion, artemisinin precludes the “secretion of tumor necrosis factor (TNF)-α, interleukin-(IL-) 1β, and IL-6”, which imparts an “anti-inflammatory effect on phorbol myristate acetate-(PMA-) induced THP-1 human monocytes” [36]. Furthermore, sesquiterpene lactones such as arteannuin B, dihydroartemisinin, artemisinic acid, and artemisinin considerably diminish “LPS-activated production of prostaglandin E2 (PGE2)” which is an inflammatory mediator bolsters tumor growth [36].

### 4.3. Antiulcer Activity

The alpha-methylene-gamma-lactone moiety of sesquiterpene lactones from *Artemisia* are responsible for the antiulcerogenic activity these compounds exhibit [51]. *A. annua*’s crude extract and the sesquiterpene lactone enriched fraction from the aerial parts displayed “antiulcerogenic activity on the indomethacin induced ulcer in rats” [36]. Three divergent polarity fractions were gleaned from the sesquiterpene lactone fraction utilized to perform column chromatography. The medium polarity fraction revealed that the active compounds of *A. annua* perform their function by elevating the “prostaglandin levels in the gastric mucosa” [36].

### 4.4. Antimicrobial Activity

The antiparasitic activity of artemisinin and its analogs displayed its effects against *Plasmodium* species both in vitro and in vivo. These compounds have proven effective against “multidrug resistant strains of the malaria parasite” and in cerebral malaria cases [36]. The antimalarial properties of artemisinin can be attributed to the endoperoxide bridge in its structure [36]. The action mechanism involves the interaction of the endoperoxide group with the endoparasitic iron which results in the establishment of free radicals [36]. The free radicals derived from artemisinin alkylate are the “malarial-specific proteins” that compromise the membranes and micro-organelles of the parasite [36]. Lastly, the radicals degrade the infected blood cells, which prompts the removal of the cell by the immune system of the host [36]. Moreover, artemisinin also targets the mitochondria of the parasite which is critical to the proper functioning of the parasite [36]. The key functional part of artemisinins and their analogues is the endoperoxidic bridge to which its antimalarial properties are attributed. The direct interaction of the endoperoxidic group of artemisinins with the intraparasitic iron results in the production of free radicals, which later damage the micro-organelles and membranes of the parasite [36]. The lifesaving drug “Artemisinin” is a secondary metabolite from *Artemisia annua*. This bioactive compound garnered clinical significance as a first in-class antimalarial drug with a clinical trial from the leaf extract of *A. annua* directed by Prof. Youyou Tu. This study was performed in August 1972 at Hainan Island on 21 patients, achieving 95–100% efficacy. It is ironic that she took the medicine herself to evaluate the safety of the extract. This ground-breaking clinical achievement led Prof. Youyou Tu to achieve a Nobel Prize in Physiology or Medicine in 2015 [52]. Recently, more attention has been paid to artemisinin’s application for treating various diseases such as antiproliferation [53], antifibrosis [54], antiviral [55], and renal protective [56]. Artemisinins have excellent anti-inflammatory and immunoregulatory functions due to its property of keeping good balance between oxidation and oxidation resistance, which is the cause of various diseases. Still, the mode of action of artemisinin is debatable among the scientific community because the detailed mechanism of action is still highly controversial [57]. The endoperoxide pharmacophore alone has stimulated the development of several different classes of endoperoxides and has been included in Figure 11.

The key extraction process involved using a cold temperature from the plant. Prof. Tu utilized ether replacing ethanol and chose leaves to obtain a concentrated artemisinin biomedicinal agent without heating. Ether with low boiling point is used in extraction processes with functional groups which are unstable by heating. The structure of artemisinin has a unique 1,2,4-trioxane peroxide pharmacophore supports this observation. Peroxides are unstable under heating conditions, but this pharmacophore was used toward development of analogs containing trioxolane functionality [58].

## 5. Conclusions

*Artemisia* species are the hardy herbaceous vegetation that are mainly grown in dry or semi-arid habitats. These plant species consist of terpenoids, phenolic compounds and flavonoids as the chief bioactive compounds, which are generally responsible for giving the plant species the characteristic bitter tastes and strong aromas. In addition, these secondary metabolites are also in charge of providing the antitumor, anti-inflammatory, anti-ulcer and anti-microbial activities that assist in promoting the health of the global population. For this particular reason, extraction, isolation and characterization of the bioactive compounds of *Artemisia* species is highly imperative.

## Figures and Tables

**Figure 1 molecules-26-06995-f001:**
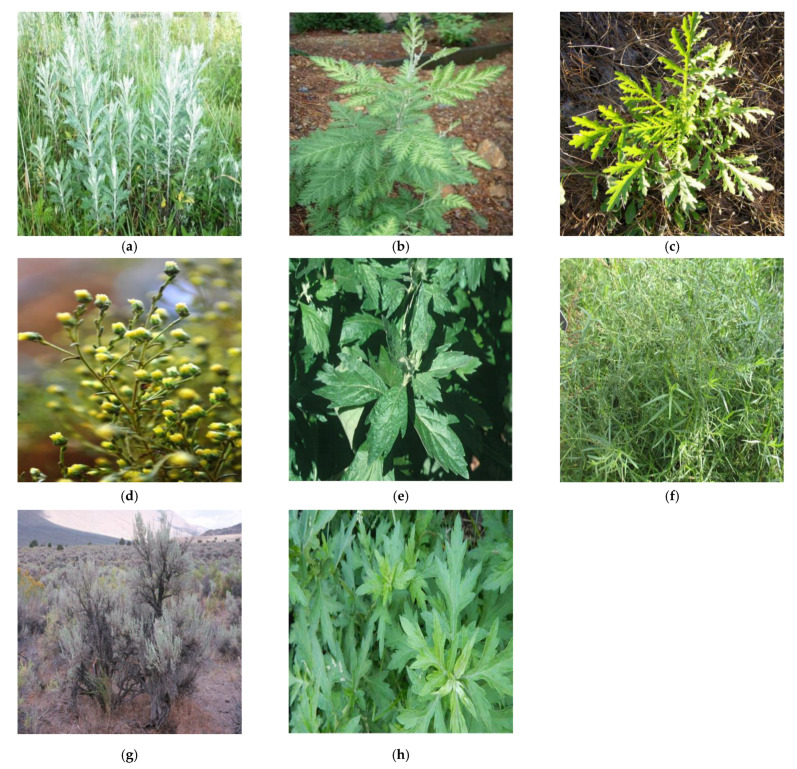
Field pictures of eight *Artemisia* species [6]: (**a**) *Artemisia* absinthium, (**b**) *Artemisia* annua, (**c**) *Artemisia* biennis, (**d**) *Artemisia* campestris, (**e**) *Artemisia* douglasiana, (**f**) *Artemisia* dracunculus, (**g**) *Artemisia tridentata*, and (**h**) *Artemisia vulgaris*.

**Figure 2 molecules-26-06995-f002:**
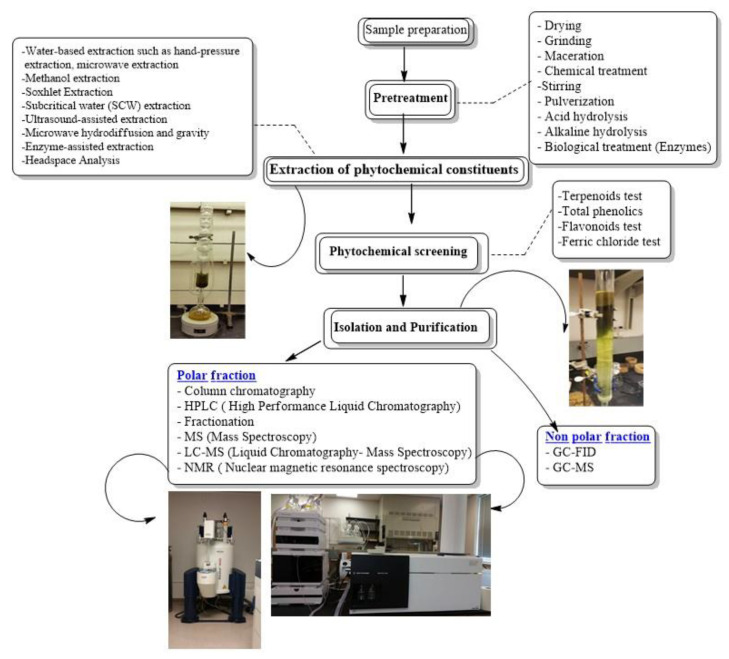
Flowchart of stages involved in extraction, isolation and characterization of bioactive compounds from *Artemisia*.

**Figure 3 molecules-26-06995-f003:**
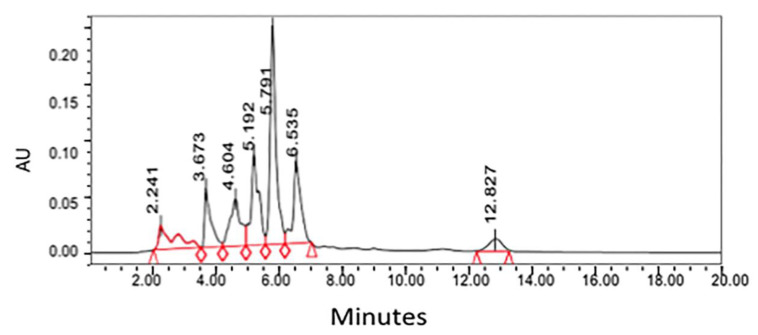
HPLC chromatogram for chloroform extract from *Artemisia* cina [39]. Peak at 5.7 represents santonin and quantified by the standard of santonin run with same HPLC condition.

**Figure 4 molecules-26-06995-f004:**
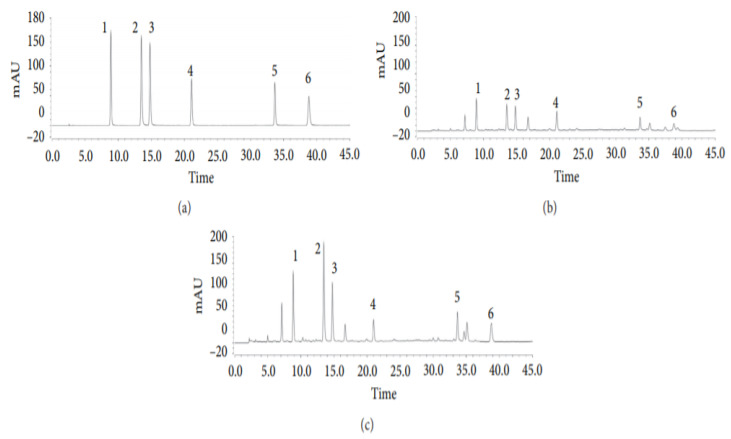
Representative HPLC chromatograms of (**a**) mixed standards; (**b**) *A. capillaris* herb; (**c**) *A. capillaris* decoction. Each numerical number signifies respective compounds: (1) 5-Caffeoylquinic acid; (2) 3-caffeoylquinic acid; (3) 4-caffeoylquinic acid; (4) 1,3-dicaffeoylquinic acid; (5) 3,4-dicaffeoylquinic acid; (6) 4,5-dicaffeoylquinic acid [40].

**Figure 5 molecules-26-06995-f005:**
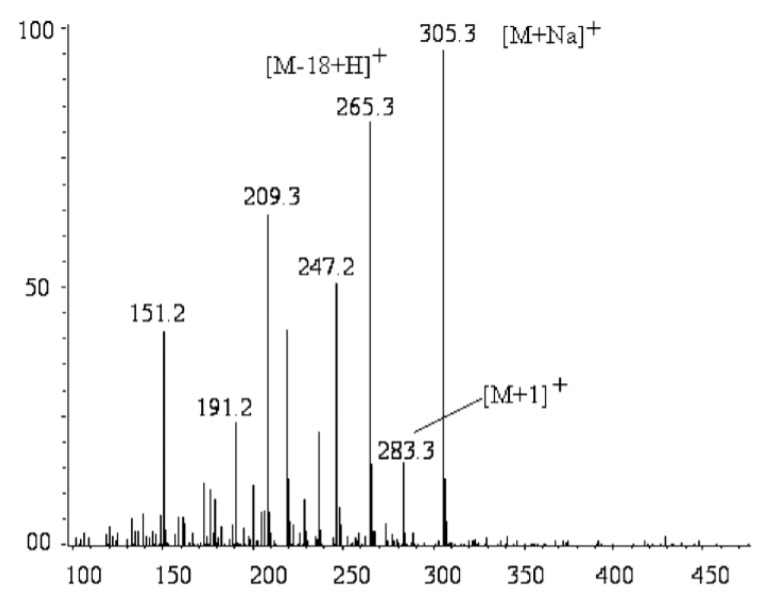
Mass spectrum of artemisinin under positive ion mode [41], where 283.3 [M + 1]^+^ represents artemisinin; 305.3 [M + Na]^+^ represents sodium adduct of artemisinin.

**Figure 6 molecules-26-06995-f006:**
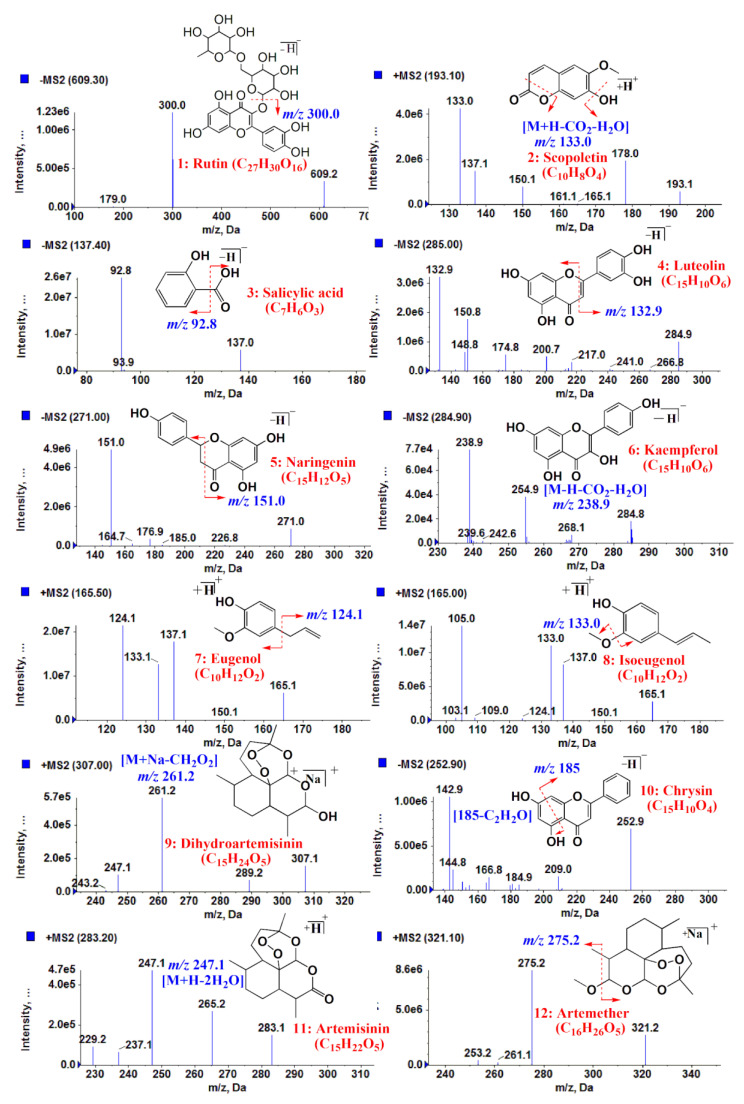
ESI-MS/MS spectra in positive and negative mode in addition to MRM transitions of various *Artemisia* species [42]. As per the report, peaks 1, 4, 5, 6 and 10, were identified as rutin, luteolin, naringenin, kaempferol and chrysin respectively. Peaks 7 and 8 were detected as eugenol and isoeugenol and peaks 9, 11 and 12 were detected as dihydroartemisinin, artemisinin and artemether, respectively.

**Figure 7 molecules-26-06995-f007:**
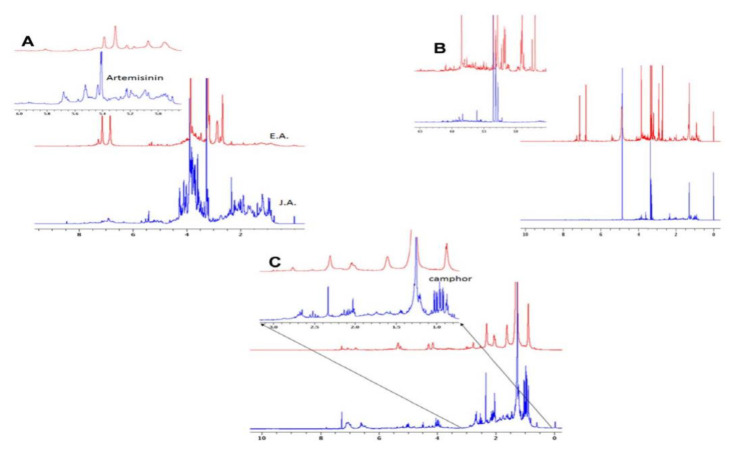
NMR analyses of the chemical compositions of the hexane–chloroform, methanol and water extracts of both EA and JA. The NMR spectrum of EA and JA water extracts (**a**). The NMR spectrum of EA and JA hexane–chloroform extracts (**b**). The NMR spectrum of the JA and EA methanol extracts (**c**). [43].

**Figure 8 molecules-26-06995-f008:**
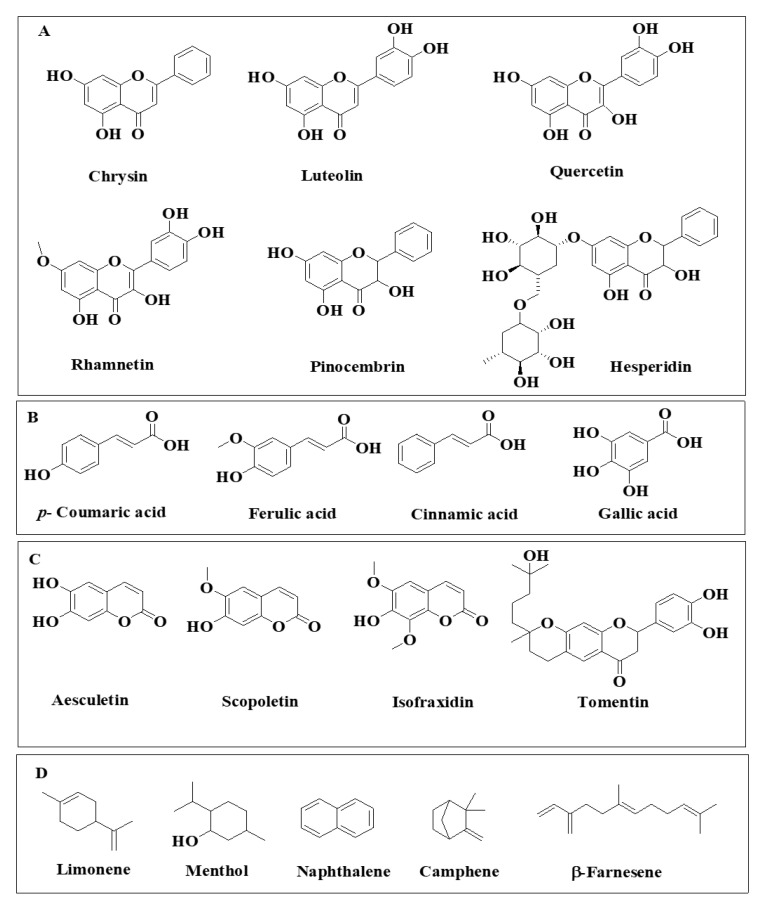
The fundamental classes of bioactive compounds present in *Artemisia*: (**A**) flavonoids; (**B**) phenolic acids; (**C**) coumarin; (**D**) terpene.

**Figure 9 molecules-26-06995-f009:**
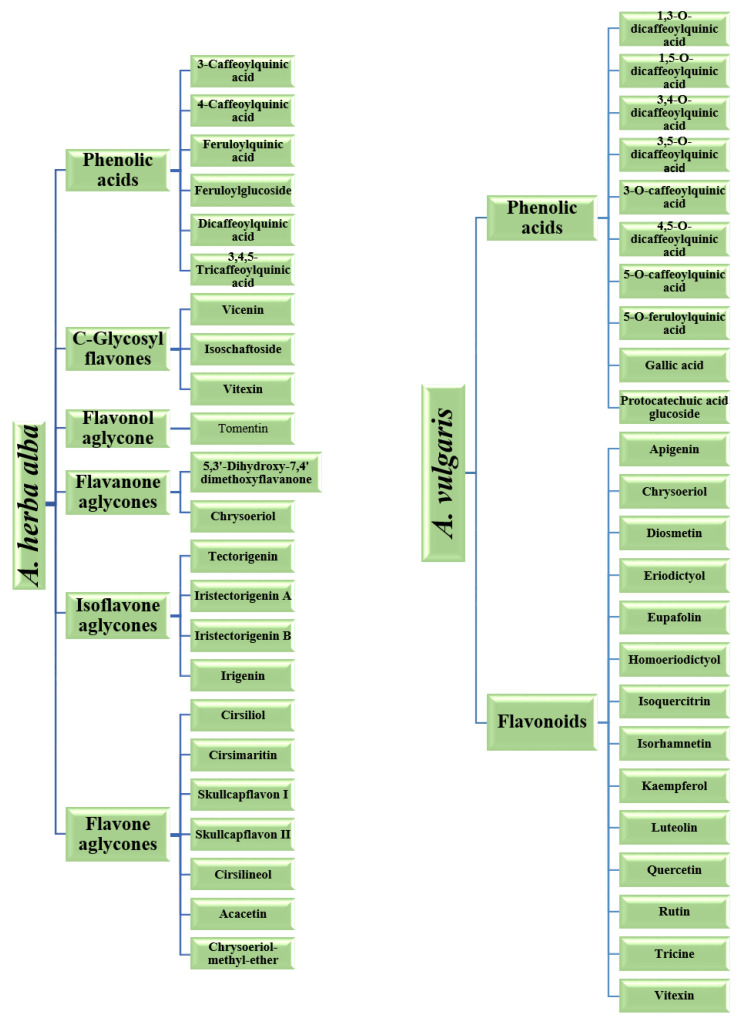
Various bioactive compounds found in *A. herba alba* [17] and *A. vulgaris* [5].

**Figure 10 molecules-26-06995-f010:**
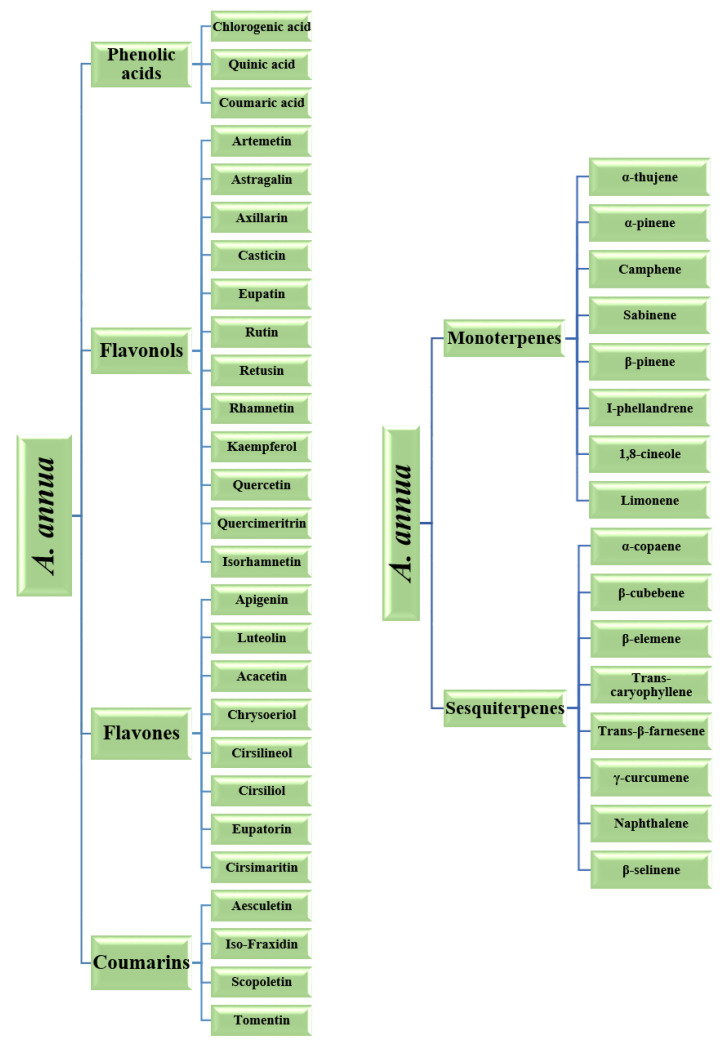
Illustration of the various bioactive compounds found in *A. annua* [17].

**Figure 11 molecules-26-06995-f011:**
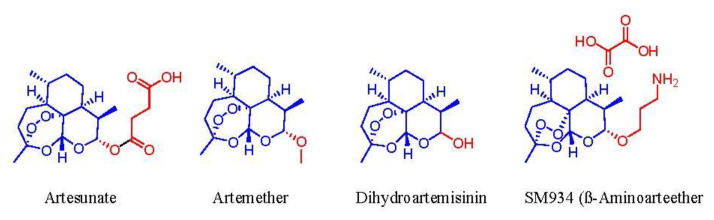
Artemisinin’s derivatives.

**Table 2 molecules-26-06995-t002:** Health bolstering effects of *Artemisia* extracts.

*Artemisia* Species Extract	Salient Applications	Animal Model and Cell Lines Used	Reference
*Artemisia* absinthium	Demonstrated reduction in the overall cholesterol level in diabetic rats through inhibition of enzyme activities involved in cholesterol biosynthesis	Alloxan-induced diabetic rats	[17,48]
*Artemisia* annua	Mice fed *A.* annua extract diet demonstrated diminished levels of malondialdehyde, a biomarker of lipid peroxidation. Studies demonstrated the use of *A.* annua extract has high anticancer effects	C57BL/6J mice Human Breast Adenocarcinoma MCF7 (BA), Human Lung Carcinoma (LC) and Chinese Hamster Ovary (CHO) cell lines and Primary Human Dermal Fibroblasts isolated from adult skin (HDFa) cells were used	[43,49]
*Artemisia* biennis	Fractions of *A.* biennis have been shown to stifle generation of reactive oxygen species (ROS) while improving the activity of superoxide dismutase (SOD).	PC12 Cells	[8]
*Artemisia* campestris	Antioxidant activity of *A.* campestris was revealed in a study that exposed rats to induced oxidative stress from a puffer fish. The rats were then fed aqueous extracts of *A.* campestris, resulting in the inhibition of thiobarbituric acid reactive substance (TBARS) and the amplification of antioxidant enzyme activities such as glutathione peroxidase (GSH-Px) and superoxide dismutase (SOD) in the brain, liver, and kidney.	Rats	[11]
*Artemisia* douglasiana	*A.* douglasiana leaf essential oil has been shown to be an efficacious complementary treatment for recurring urinary tract infection.	Studied in rodents	[14]
*Artemisia* dracunculus	*A.* dracunculus is typically used to enhance a poorly functioning digestive system by improving appetite to remove toxins from the body. In cultures with a particularly high intake of red meat, *A.* dracunculus is employed as a digestive stimulant.	Phenylbutazone-induced ulcer in rats	[14]
*Artemisia* tridentata	*A.* tridentata possesses anthelmintic activity which is useful in the treatment of oxyuriasis and ascariasis by inducing paralysis on the worm to flush out the parasite from the bowls.	-	[17]
*Artemisia* vulgaris	The antihypertensive activity of *A.* vulgaris has been demonstrated to limit the hypertensive effects of noradrenaline Cytotoxic activity inhibits growth of tumor cells in cancer cell lines	Isolated perfused rat mesentery SW-480, MCF7, HL-60, HeLa, 293T, and A7R5.	[32]

## Data Availability

Not available.

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
