# Peer review of "Extraction, Isolation and Characterization of Bioactive Compounds from Artemisia and Their Biological Significance: A Review"

_molecules, 2021, doi:10.3390/molecules26226995_

Round 1

Reviewer 1 Report

  1. Pictures should all have the same size in figure 1.
  2. Flow chart legend needs more details. Images are NOT sharp enough.
  3. Figure 3 also more information in figure legend is missing.
  4. The same in figure 5.
  5. Figure 6, the same, also figures are nor sharp.
  6. In 4.2: 

    NF-?B is a well-known protein that functions to modulate 150+ “inflammatory genes and mediates the immune 585 response in humans” 

    More information concerning NF-?B is required, since this factor is not just a protein.
  7. In 4.4:

    "Moreover, artemisinin also targets the mitochondria of the parasite which is critical to the proper functioning of the parasite"

    Please give more detail about this fact.
  8. A conclusion is missing.

Author Response

We thank the reviewer for valuable comments and suggestions. Please find answers to all comments laid by the reviewer. 

  1. Pictures should all have the same size in figure 1.

Answer: All pictures in figure 1 have been adjusted.

  1. Flow chart legend needs more details. Images are NOT sharp enough.

Answer: Flow chart has been modified with more details.

  1. Figure 3 also more information in figure legend is missing.

Answer: More information has been added to the legend.

  1. The same in figure 5.

Answer: Information has been added.

  1. Figure 6, the same, also figures are nor sharp.

Answer: Information has been added.

  1. In 4.2: 

NF-?B is a well-known protein that functions to modulate 150+ “inflammatory genes and mediates the immune 585 response in humans” 

More information concerning NF-?B is required, since this factor is not just a protein.

Answer: More details have been included in the revised manuscript.

  1. In 4.4:

"Moreover, artemisinin also targets the mitochondria of the parasite which is critical to the proper functioning of the parasite"

Please give more detail about this fact.

Answer: More details have been added in the revised manuscript.

  1. A conclusion is missing.

Answer: Conclusion has been added.

Reviewer 2 Report

Most Artemisia species are traditional herbal medicines around the world. There are thousands literatures concerning the chemical constituents of Artemisia plants and their bioactivities. This review was made on only 51 literatures, some literatures' availability is poor and even not correctly indexed in the context. The plants' scientific names were not printed correctly in the literature list. Artemisia annua is called scientifically "huanghuahao"(yellow flower Artemisia) not "qinghao" in Chinese, and it distributes from north to south almost all part of China, not just north of Suiyan and Chatar. And Suiyan and Chatar were ancient province name, nowaday part of Inner Mongolia and Hebei provinces. Artemisinin from Artemisia annua is a good drug to cure malaria, because of it, TU Youyou received Nobel prize on behave of hundreds of Chinese scientists. They published research results in Chinese journals in last century and in the Lancet, New England Journal of Medicine and Nature Medicine in the beginning of this century, which this review did not mention.

Author Response

We thank the reviewer for valuable comments and suggestions. 

Answer: All the references has corrected as per the written text. Correction has been made for “Artemisia annua “regarding it name and distribution on the geographical map. The published results Artemisinin from Artemisia annua has been added in the revised manuscript. A paragraph has been added related to the discovery of Artemisinin and 2015 Nobel prize.  

Reviewer 3 Report

The review is devoted to an interesting subject and well organized. However, the authors must consider several issues as follows:

  1. A general problem that persists throughout the manuscript is the mismatch between text and bibliography, for example: in subchapter 1.1 at Artemisia annua, references [12] and [14] are not appropriate within the text; in subchapter 1.1 at Artemisia biennis (lines 103-122) the related bibliography seems to be [11-14] instead of [15-17]; the same for Artemisia campestris instead of [18] should probably be [15] (lines 125, 126, 129, 132); for Artemisia douglasiana instead of [19] it should probably be [17] (lines 144), etc. The same applies to the other subchapters: [35] to be replaced by [37] (lines 283, 284), [36] by [38] (lines 336,339,348, 351,354), [38] by [40] (lines 358,359,364,365), and so on. I would ask the authors to revise the entire manuscript and the related bibliography and to make the necessary corrections.
  1. Please delete superscript “3” (line 343).
  2. Please introduce the correct text in Figure 4 as follows: “Representative HPLC chromatograms of: a. mixed standards” (line 456).
  3. In Chapter 3. Bioactive compounds from Artemisia I would suggest the introduction of a subchapter in which the authors also discuss Coumarins.
  4. The therapeutic applications/properties of the Artemisia species presented in table 2 should be better clarified in accordance with what is already presented in the introduction and to be underlined in each case where they were observed (human, experimental animal/cell line type).
  5. Figures 9 and 10 illustrate various bioactive compounds only from the species A. annua, A. herba alba and A. vulgaris. Considering that in the introduction part it was reported the existence of bioactive compounds in other Artemisia species (for example A. campestris - lines 130-132, A. dracunculus - lines 157-159, A tridentate - lines 169-170) it would be recommended to render similarly to Figures 9 and 10 and the bioactive compounds from these Artemisia species.
  6. Reference [53] - line 596 does not exist in the bibliography at the end of the manuscript.
  7. A slightly more comprehensive bibliography would be welcome.

Author Response

We thank the reviewer for valuable comments and suggestions. Please find answers to all comments laid by the reviewer. 

  1. A general problem that persists throughout the manuscript is the mismatch between text and bibliography, for example: in subchapter 1.1 at Artemisia annua, references [12] and [14] are not appropriate within the text; in subchapter 1.1 at Artemisia biennis (lines 103-122) the related bibliography seems to be [11-14] instead of [15-17]; the same for Artemisia campestris instead of [18] should probably be [15] (lines 125, 126, 129, 132); for Artemisia douglasiana instead of [19] it should probably be [17] (lines 144), etc. The same applies to the other subchapters: [35] to be replaced by [37] (lines 283, 284), [36] by [38] (lines 336,339,348, 351,354), [38] by [40] (lines 358,359,364,365), and so on. I would ask the authors to revise the entire manuscript and the related bibliography and to make the necessary corrections.

Answer: All the bibliography has been modified as per the text.

  1. Please delete superscript “3” (line 343).

Answer: Has been deleted. 

  1. Please introduce the correct text in Figure 4 as follows: “Representative HPLC chromatograms of: a. mixed standards” (line 456).

Answer: Figure 4 has been corrected as per suggestion.

  1. In Chapter 3. Bioactive compounds from Artemisia I would suggest the introduction of a subchapter in which the authors also discuss Coumarins.

Answer: Coumarins has been added and discussed briefly.

  1. The therapeutic applications/properties of the Artemisia species presented in table 2 should be better clarified in accordance with what is already presented in the introduction and to be underlined in each case where they were observed (human, experimental animal/cell line type).

Answer: In the revise manuscript, table 2 have been modified as per the suggestion.

  1. Figures 9 and 10 illustrate various bioactive compounds only from the species A. annuaA. herba alba and A. vulgaris. Considering that in the introduction part it was reported the existence of bioactive compounds in other Artemisia species (for example A. campestris - lines 130-132, A. dracunculus - lines 157-159, A tridentate - lines 169-170) it would be recommended to render similarly to Figures 9 and 10 and the bioactive compounds from these Artemisia species.

Answer: Figure 9 and 10 has been corrected with new reference.

  1. Reference [53] - line 596 does not exist in the bibliography at the end of the manuscript.

Answer: All the references have been arranged correctly.

  1. A slightly more comprehensive bibliography would be welcome.

Answer: Bibliography has been modified.

Reviewer 4 Report

The article by Sharma et al. reviewed the extraction, isolation, and characterization methodologies of active compounds from various Artemisia species, which demonstrates that divergent species of Artemisia exhibit a vast array of biological activities such as anti-malarial, antitumor, and anti-inflammatory activities. This paper was well organized and can be published in Molecules after taking the following points into consideration:

  1. Why the “Percent Yield” number in the first line of Table 1 was much higher than other values?
  2. Under the background of green chemistry and sustainable development, there should be reports on the extraction of active components from Artemisia species using green solvents. The authors should mention such kind of reports in this review.
  3. The authors should apply for permissions from the publishers if the figures were adapted from literature.
  4. Conclusions and perspective should be given at the end of this article.

Author Response

We thank the reviewer for valuable comments and suggestions. Please find answers to all comments laid by the reviewer

  1. Why the “Percent Yield” number in the first line of Table 1 was much higher than other values?

Answer: The percent yield was a mistake and has been removed for the consistency of the table.

  1. Under the background of green chemistry and sustainable development, there should be reports on the extraction of active components from Artemisia species using green solvents. The authors should mention such reports in this review.

Answer: Thank you for pointing out the green chemistry and sustainable chemistry. The reference 33 is all about the green extraction of bioactive compounds from artemisia.

  1. The authors should apply for permissions from the publishers if the figures were adapted from literature.

Answer: Copyright has been submitted

  1. Conclusions and perspective should be given at the end of this article.

Answer: Conclusion has been included.